# Monitoring and responding to emerging infectious diseases in a university setting: A case study using COVID-19

**K. James Soda** [1] *, **Xi Chen**[2], **Richard Feinn**[3], **David R. Hill**[3]

1 Department of Mathematics and Statistics, Quinnipiac University, Hamden, Connecticut, United States of America, 2 Department of Sociology and Anthropology, Quinnipiac University, Hamden, Connecticut, United States of America, 3 Department of Medical Sciences, Frank H. Netter MD School of Medicine, Quinnipiac University, Hamden, Connecticut, United States of America

* Kenneth.Soda@quinnipiac.edu

**Data Availability Statement:** The version of the Covasim model described and used in this paper is available at https://github.com/kjamessoda/covasim.git in the F20ModelAndTesting branch. All

## Abstract

Emerging infection diseases (EIDs) are an increasing threat to global public health, especially when the disease is newly emerging. Institutions of higher education (IHEs) are particularly vulnerable to EIDs because student populations frequently share high-density residences and strongly mix with local and distant populations. In fall 2020, IHEs responded to a novel EID, COVID-19. Here, we describe Quinnipiac University's response to SARS-CoV-2 and evaluate its effectiveness through empirical data and model results. Using an agent-based model to approximate disease dynamics in the student body, the University established a policy of dedensification, universal masking, surveillance testing via a targeted sampling design, and app-based symptom monitoring. After an extended period of low incidence, the infection rate grew through October, likely due to growing incidence rates in the surrounding community. A super-spreader event at the end of October caused a spike in cases in November. Student violations of the University's policies contributed to this event, but lax adherence to state health laws in the community may have also contributed. The model results further suggest that the infection rate was sensitive to the rate of imported infections and was disproportionately impacted by non-residential students, a result supported by the observed data. Collectively, this suggests that campus-community interactions play a major role in campus disease dynamics. Further model results suggest that app-based symptom monitoring may have been an important regulator of the University's incidence, likely because it quarantined infectious students without necessitating test results. Targeted sampling had no substantial advantages over simple random sampling when the model incorporated contact tracing and app-based symptom monitoring but reduced the upper boundary on 90% prediction intervals for cumulative infections when either was removed. Thus, targeted sampling designs for surveillance testing may mitigate worst-case outcomes when other interventions are less effective. The results' implications for future EIDs are discussed.

other relevant data and code are within the paper and its Supporting Information files.

**Funding:** KJS and XC received Grant-in-Aid funding from the Quinnipiac University College of Arts and Sciences (https://www.qu.edu/schools/arts-and-sciences/), and XC received funding from the Quinnipiac University Provost Office's Faculty Scholarship and Creative Works Impact Fund (https://www.qu.edu/quinnipiac-today/provosts-fall-2022-update-2022-08-23/#learning). These funds covered publication fees. Both internal funding sources do not receive a grant number. The funders had no role in study design, data collection and analysis, decision to publish, or preparation of the manuscript.

**Competing interests:** The authors have declared that no competing interests exist.

# Introduction

Emerging infectious diseases (EIDs) increasingly pose a threat to global public health. An EID is any pathogenic disease that has been newly introduced to an area or whose incidence has rapidly increased [1, 2]. Even after correcting for differences in sampling effort, EIDs have originated at an increasing rate since 1940 [3]. Modern trends in global connectivity, population growth, and interactions at the human-environment interface are all theorized to contribute to the growth of EIDs by accelerating the rate at which local outbreaks can propagate globally [4, 5]. Novel diseases are an especially problematic subcategory of EIDs because there is no existing literature for these diseases on which public health officials can inform their responses. To reflect the threat of novel pathogens, the World Health Organization has placed Disease X on its list of priority diseases for research and public health planning since 2015 [6]. The Disease X designation is a placeholder for a yet unknown pathogen with the capacity to cause a global pandemic, and is intended to encourage the scientific and public health communities to develop response plans that can apply to non-specific pathogens [7]. The Disease X concept became a reality in 2020 as the novel SARS-CoV-2 virus led to the ongoing COVID-19 pandemic. In light of concerns that new strains of coronaviruses or similar respiratory infections (*e.g.*, influenza) could cause the next global pandemic [5, 8], it is critical that the public health community take stock of what measures were most effective in early COVID-19 responses in order to prepare for the next EID or Disease X.

Institutions of higher education (IHEs) pose unique public health challenges in the face of EIDs with droplet and airborne transmission, in part due to how most university-owned properties are structured. Most university-owned residences feature high densities of students living in individual rooms with high densities of rooms on each floor, which can accelerate disease transmission [9, 10]. Further, lavatory facilities in many university residences are shared between several rooms. This poses an additional challenge because basic hygiene practices, such as bathing and dental care, preclude masking, and surfaces such as faucets and door handles facilitate fomite transmission when applicable. Finally, transmission is likely to jump between university residences as their occupants intermingle at shared facilities (e.g., food services, athletic facilities, libraries) and in the classroom.

Further, university-owned residences are differentiated from other residential facilities (*e.g.*, hospitals, long-term care facilities, prisons) through the multiscale population mixing that is inherent to campus communities. On a local level, university students regularly mix with surrounding communities through student employment, peers in off-campus housing, interactions with friends and family outside of the university, and participation in community events. Additionally, most IHEs include students originally from populations beyond those surrounding the campus. This means that student populations can be expected to mix at intra- and international scales, especially at the beginning and end of semesters and following major holidays. This increased mixing can pose a double threat to infection containment. On the one hand, mixing could cause infections to spill over from IHEs into connected populations. On the other hand, mixing can introduce infections into an IHE's population, where the previously discussed factors can amplify disease transmission.

Quinnipiac University is a suburban university located in New Haven County, Connecticut, USA. In the fall 2020, the University had 6,841 undergraduate students and 2,903 graduate students [11]. The University allowed students to attend classes via an online-only or hybrid (*i.e.*, partially in-person and partially online) modality. Approximately 7,100 students selected the hybrid modality. Roughly 55% of undergraduates lived on campus [11]. University-owned residences included 15 dormitories, 10 townhouses, and 53 free-standing houses that ranged in size from two to four bedrooms.

In preparation for the 2020/2021 academic year, the University established a COVID-19 task force charged with developing a plan to allow in-person and remote instruction while minimizing the risk of COVID-19 transmission. Like other institutions (*e.g.*, [12–15]), Quinnipiac de-densified classrooms to allow physical distancing and required facemask usage in most public spaces. The maximum capacity in university residences was reduced to no more than two students per room. To facilitate remote learning, classrooms were equipped with specialized audio-visual equipment, and professors at high risk for severe illness were allowed to teach online. The task force's plan also emphasized student testing protocols to detect COVID-19 cases, and instituted isolation, quarantine, and contact tracing protocols upon case identification. The testing protocols used both random surveillance testing and symptom-based monitoring. To assist symptom-based monitoring, the University distributed a smart-phone application that logged the user's COVID-related symptoms each day and instructed the user to seek testing when their symptoms suggested COVID-19 infection. The COVID-19 protocols required additional, fine-grained decisions, such as the number of rooms needed for isolation and quarantine, the number of contact tracers required to follow transmission chains, and the optimal strategy for sampling students for surveillance testing. Because SARS-CoV-2 was a Disease X scenario, there was little data to inform these decisions. To address this knowledge gap, the task force formed a subcommittee to develop a model of how COVID-19 could spread through the student body during the fall semester based on available knowledge about SARS-CoV-2 transmission, contact patterns in college-aged individuals, and the spatial structure of university-owned residences. The modeling team worked on the premise that not all students needed to be tested to reliably identify COVID-19 circulation amongst the student body.

The modeling subcommittee used an open-source agent-based model for community COVID-19 dynamics, Covasim, as a basis for their model, but modified Covasim to better reflect the nested housing structure found in university-owned residences [16]. An agent-based model simulates a system using computational units, called agents, that interact with each other based on well-defined rules and seeks to explore what patterns arise in the simulation that are not immediately predictable based on the agents' individual actions [17]. Agent-based models are well-suited to inform policy decisions because they provide a granular depiction of their populations and explicitly simulate the behavior of individuals. As a result, individual-based interventions, such as contact tracing, and traits, such as contact networks, are more easily translated into the model. Covasim has been used to guide primary and secondary school reopening strategies in the United States and the United Kingdom [18–21] and was similarly adapted to guide public health policy at Boston University [22]. However, this was the first use of the adapted form developed at Quinnipiac.

In this paper, we detail what modifications we made to Covasim to better simulate a university's student body in general and Quinnipiac University in particular. We also describe Quinnipiac's COVID-19 testing protocol, particularly the survey methodology design used to sample students for surveillance testing. Finally, we compare the performance of the model against the actual case numbers during the fall semester of 2020, evaluate the effectiveness of the University's protocols, and discuss its implications for university responses to future Disease X scenarios.

## Materials and methods

### Covasim model

Our model extended the Institute for Disease Modeling's Covasim model [16]. Covasim is a stochastic, agent-based model of COVID-19 dynamics that assigns one agent to each

individual or set of identical individuals. In our implementation, each agent represented one student. The model builds on a discrete-time susceptible-exposed-infectious-recovered model, but its agent-based structure allows it to explicitly incorporate relevant interhost properties, such as viral loads, age-dependent susceptibility and transmissibility, and heterogeneity in susceptibility, transmissibility, and recovery times. The model also organizes agents into sets of pools, each representing a different social context (*e.g.*, home, work, and school). On each day, every individual forms a new contact network by sampling a Poisson-distributed number of agents from its pools, thus allowing the model to simulate heterogeneous population mixing through pool membership. Disease transmission randomly occurs when an infectious individual contacts a susceptible individual. The probability of transmission depends on the pool associated with the contact and on the current properties of each agent (*e.g.*, infectious host's viral load). Covasim assigns an outcome to each infection, such as whether symptoms develop, and every infection may be diagnosed through testing. The version of the Covasim model that we used, including the added features discussed here, are available through GitHub (https://github.com/kjamessoda/covasim.git) and within the F20ModelAndTesting branch.

## Contact network structure

We modified Covasim's contact pools so that the pools better emulated the structure of university-owned residences. Contact pools focused on housing structure rather than classroom structure because we assumed that the University's physical distancing and masking policies would lower transmission rates within the classroom to a negligible level. This assumption was later confirmed through contact tracing data, and has been borne out in other university settings [23]. Contact pools fell into four categories: community, floor, bathroom, and room. Every agent belonged to a single community pool. Then, subsets of the community pool formed floor pools, each representing students living on the same floor of a dormitory or in a comparable unit of a university-owned residence. In turn, subsets within each floor pool formed bathroom pools, each representing students who shared the same bathroom, and subsets within the bathroom pools formed room pools, each representing a single set of roommates. A single room pool represented students living in free-standing houses. The mean number of contacts drawn from each pool to create a contact network could differ between pools (see Simulation Setup). Room pools were the exception to this rule; each agent's contact network contained their roommates every day, an assumption that both the University's contact tracing data and reports from other universities subsequently validated [9].

To account for students living outside of university-owned housing, agents were divided into residential students and non-residential students. Non-residential students at Quinnipiac lived in diverse housing arrangements (*e.g.*, cohabitating with other students, living with family members), so we did not represent their contact networks using floor, bathroom, and room pools. The community pool, however, functioned in the same manner for both student types.

## Imported infections

To represent disease transmission from non-students to students, the model randomly selected agents to potentially transition to the exposed compartment without contacting an infectious student (*i.e.*, an imported infection). Although exposed, infected, and recovered agents were eligible for selection, only susceptible agents transitioned. Residential and non-residential students were sampled separately each day, and the number of agents sampled was a Poisson random variable whose mean was proportional to the size of the agent group, the expected number of non-student contacts per student, and the estimated prevalence of COVID in Connecticut (see Simulation Setup; S1 Appendix).

## Conversion from $R_0$ to probability of transmission

Covasim does not explicitly use the basic reproductive number ($R_0$) as a model parameter. The closest analog is the base probability of transmission ($\beta$). Nonetheless, it is possible to extend Covasim's assumptions to relate an *a priori* value for $R_0$ to $\beta$ (S1 Appendix). $R_0$ is defined as the expected number of new infections in a fully susceptible, well-mixed population after a single infectious individual is introduced [24]. If the age structure of the population emulates that of the United States [25] (though see Table A in S1 Appendix for a minor deviation), then Covasim's models of viral load dynamics and disease transmission imply:

$$\beta \approx \frac{R_0}{8.093758 n_c}$$

where $n_c$ is the expected number of contacts one individual has in one day.

## Interventions

Covasim provides testing-based public health interventions and contact tracing interventions [16]. Testing-based interventions receive a set of agents and change infected agents to the diagnosed state based on a provided test sensitivity. We set the sensitivity to 0.95 to emulate nasal swab PCR tests for SARS-CoV-2 virus [26]. The model had a one-day delay between testing and diagnosis. Upon diagnosis, individuals were transferred to isolated populations and made no further contacts until entering the recovered compartment. Each diagnosis also led to a reported case in the model's output. Testing-based interventions differed in how the sets of individuals to test were generated.

We implemented three testing-based interventions in our simulations, each corresponding to one of the University's public health interventions. In fall 2020, every member of Quinnipiac University, including students, was asked to complete a daily public health application that screened for COVID-19 symptoms (MyOwnMed COVID-19 symptom application; MyOwnMed, Inc. Bethesda, Maryland 20817). Students with symptoms were asked to contact Student Health Services, be evaluated, receive a COVID-19 test, and isolate until the results returned. Student Health Services also monitored the symptom application dashboard and contacted students who reported COVID-19-like symptoms for further evaluation. To simulate the application's impact, each day a symptom-monitoring intervention identified symptomatic agents in the simulation who had not been tested for COVID-19. The intervention transferred these students to the isolated population and tested them using the procedures above. If the student tested negative for SARS-CoV-2, they returned to the general population after the one-day delay. To account for imperfect app usage, we set the daily probability of detection to 0.5, corresponding to an expected one-day delay between symptom onset and detection. In addition to symptom-monitoring, the University randomly tested residential students based on a targeted sampling design and non-residential students using a simple random sample (SRS) (see Sampling Strategy). Since the model integrated the structure of university residences into its contact networks and differentiated students by housing status, we designed a second testing-based intervention to replicate the targeted design for residential students and a third to replicate the SRS for non-residential students.

Contact tracing interventions were carried out by university contact tracers. To reflect this, after a testing-based intervention moved an infectious agent into isolation, a contact-tracing intervention retrieved the infectious individual's contact network and quarantined its contacts to the isolated population. In our implementation, contact tracing always successfully identified roommate contacts, but all remaining contacts had a 0.75 probability of identification. To be cautious, there was a two-day delay within the simulation between when contact tracing

started and when identified contacts moved to isolation; however, real tracing usually required less time. Quarantined individuals reentered the general population after 14 days if they tested negative for SARS-CoV-2 virus and after entering the recovered compartment if they tested positive.

## Simulation setup

We simulated COVID-19 dynamics in the University's student body between Aug. 31 and Nov. 24, 2020, the period between the beginning of classes and the Thanksgiving holiday; nearly all university students remained at home following Thanksgiving. Every simulation contained 3,636 residential students, each associated with a university property through their contact network structure (see Contact Network Structure), and 3,789 non-residential students, leading to a total population of 7,425 students. These population sizes slightly differ from the actual population sizes in fall 2020, which were unavailable during the summer. Every agent was randomly assigned an age between 18 and 22. Each simulation implemented the four interventions above (*i.e.*, app-based symptom monitoring, targeted surveillance testing of residential students, simple random surveillance testing of non-residential students, and contact tracing interventions ran daily. We estimated Connecticut's COVID-19 prevalence to surveillance testing schedule, the surveillance testing interventions ran on every simulated Tuesday and Wednesday. The symptom monitoring and contact tracing interventions ran daily. We estimated Connecticut's COVID-19 prevalence to be roughly 2.8 infectious individuals per 1,000 residents based on the statewide case incidence between Aug. 30 and Sept. 2, 2020 (S1 Appendix). This estimate informed the rate of imported infections in the simulations.

We simulated four main scenarios, each pairing one of two $R_0$ values (see Conversion from $R_0$ to Probability of Transmission), 1.5 or 2.5, and one of two average contact rates, 8 contacts/day or 10 contacts/day. The $R_0$ value of 2.5 was selected based on the best estimate provided in the Centers for Disease Control and Prevention's COVID-19 Pandemic Planning Scenarios, as published on May 20, 2020 [24]. Li *et al.* [27] arrived at a similar estimate for $R_0$ (2.2) based on case data from December 10, 2019 to January 4, 2020 in Wuhan, China. However, since Covasim incorporated $R_0$ through the base probability of transmission and preventive measures, such as social distancing and masking, also partly entered the model through this same probability, we also incorporated scenarios where $R_0$ was 1.5 to account for these interventions. This value agreed with one of the effective reproductive numbers used in Paltiel, Zheng, and Walensky [28]. Later model-based estimates for effective reproductive numbers in the United States during the summer of 2020 retroactively supported this choice [29]. Bharti *et al.* [30] informed the scenarios' contact rates through a preprint that Bharti and colleagues provided to us prior to their paper's official publication. Table 1 lists how the contacts were allocated across pools. Since non-residential students only belonged to the community pool, we allocated their remaining contacts to non-students under the assumption that non-residential students would mix more with the surrounding community. The rate of imported infections under each scenario was established based on the scenario's probability of transmission and average contact rate (Table 2 and S1 Appendix). Every scenario was run 1,000 times to generate a distribution of possible outcomes. The median outcome on each day was used as a prediction, and the 5th and 95th percentiles provided a 90% prediction interval.

Although it did not inform policy, we also assessed model sensitivity to the rate of imported infections by rerunning each main scenario with half and twice the rates provided in Table 2. Since there were initially no infected individuals in the simulations, every transmission chain must begin with an imported infection. Assuming that the expected number of infections in every transmission chain is constant given residential versus non-residential designation and

**Table 1. Average number of contacts per day from each pool under two total contact rates.**

| Pool | 8 contacts/day | | 10 contacts/day | |
|---|---|---|---|---|
| | Residential | Non-Residential | Residential | Non-Residential |
| Community | 2.5 | 2.5 | 4 | 4 |
| Floor | 2 | - | 2 | - |
| Bathroom | 3 | - | 3 | - |
| Room | Full Pool | - | Full Pool | - |
| Non-Student | 0.5 | 5.5 | 1 | 6 |

that the susceptible pool does not become significantly depleted, the expected number of infections at the end of a simulation should be proportional to the rate of imported infections. We therefore measured the model's sensitivity to the rate of imported infections as the scaling constant on this relationship as estimated using a least-squares line through the median predictions that is constrained to pass through the origin. Least-squares lines were fit using routines in the SciPy Python library [31, 32]. Python code to implement each scenario is available in S1 File.

## Sampling strategy for surveillance testing

To minimize the risk of initial spread, every student completed a PCR test for SARS-CoV-2 prior to coming to campus in August 2020 and again within two weeks of campus arrival. Subsequently, students were randomly sampled for testing. The student body was divided into four categories: residential students, non-residential undergraduates, non-residential graduate students, and student athletes (S1 Appendix).

Most Quinnipiac students were residential students. For these students, we applied a targeted sampling design that combined strict stratified and cluster sampling methods. We defined building floors and off-campus houses as strata and determined the number of students to select from each stratum. Guided by model results, the sampling rate for each stratum was initially 15%. The choice of 15% gave the university the highest likelihood of detecting an outbreak, without having to test all students. Although a stratified sampling strategy would reduce the standard error on any resulting incidence estimate, our goal was to increase the likelihood of detecting an outbreak through even sampling coverage across floors, suites, and houses, rather than estimating epidemiological parameters. After establishing the sample size for each stratum, we used a cluster sampling method to randomly select students to test. Each cluster was a dorm room or suite. First, SRS selected the appropriate number of dorm rooms or suites for each stratum; then one student was sampled from each selected dorm room and suite through SRS. Such an approach maximized the number of dorm rooms and suites being selected.

We used more traditional sampling designs for the remaining three student sub-populations. The sampling rate for student athletes was initially 80% and was stratified by team (*e.g.*,

**Table 2. The rate of imported infections (in infections/week) under four scenarios with different basic reproductive numbers ($R_0$) and average contact rates.**

| | | | Average Contact Rate (contacts/week) | |
|---|---|---|---|---|
| | | | *8* | *10* |
| Residential | $R_0$ | *1.5* | 0.8279664 | 1.324746 |
| | | *2.5* | 1.379944 | 2.207910 |
| Non-Residential | $R_0$ | *1.5* | 9.490872 | 8.282943 |
| | | *2.5* | 15.81812 | 13.80490 |

men's hockey, woman hockey, *etc.*). Non-residential undergraduate and graduate students were generally selected via SRS and initially at 25% and 15%, respectively, although each week's sample had to contain at least one student from every address that housed three or more students. As the semester progressed, we adapted each subpopulation's sampling rate in response to trends in the observed case incidences (Table B in S1 Appendix).

## Comparison of sampling strategies

To assess the efficacy of our targeted sampling design for surveillance testing, we compared our model's predicted infection trajectory under the targeted design to an SRS design and to complete sampling. To make the SRS design more comparable to the targeted design, the SRS scenarios randomly sampled 355 students on Tuesdays and 225 students on Wednesdays. Due to rounding error at each stratum and the desire to split surveillance testing across two days, the targeted design used these same sample sizes before accounting for individuals in quarantine or isolation. The complete sampling scenarios evenly split surveillance tests between Tuesdays and Wednesdays. We compared these three strategies under the four main scenarios described in Simulation Setup.

To further explore the relationship between sampling designs for surveillance testing and other public health interventions, we also ran holdout scenarios where contact tracing or app-based symptom monitoring were withheld. Since targeted sampling was only used on residential students, each holdout scenario solely simulated residential students. We also ran a baseline all-interventions scenario where all interventions were used to assess the impact of removing non-residential students from the simulation. There were two holdout scenarios for app-based symptom monitoring. In the app-based symptom monitoring intervention, the expected waiting time between when an individual developed symptoms and when the individual sought testing was one day to reflect the application's impact. To assess how each sampling design might have performed in the absence of app-based monitoring, we changed the expected waiting time to 4.82 days, the estimated mean waiting time between symptom onset and first clinical visit in Khalili *et al.* [33]. We called this the delayed-symptom-testing scenario. As an upper extreme, we also ran a scenario with no symptom-based monitoring (*i.e.*, all testing was surveillance testing). In all holdout scenarios, $R_0$ was 2.5, and the average contact rate was 10 contacts/day. These holdout scenarios also provided guidance on the impact of individual interventions and the role of non-residential students in disease transmission.

As with the main scenarios, 1,000 simulations comprised each holdout scenario. We assessed the differences between holdout scenarios based on their median cumulative infections across simulations and on their 90% prediction intervals (*i.e.*, 5th and 95th percentiles). Python code to implement each sampling-strategy and holdout scenario is available in S1 File.

## Ethics statement

Quinnipiac University's Institutional Review Board determined that the surveillance sampling design detailed here fell under the category of public health surveillance and not research, and thus, the design did not require further board consideration. All case incidence data reported here was used retrospectively and was fully anonymized before the researchers accessed it. Therefore, this data does not fulfill the US Office for Human Research's definition of human subject data.

## Results

### Adjustments to sampling strategy

Throughout the fall semester, we adjusted the proportion of students sampled in each category (*i.e.*, residential undergraduates, non-residential undergraduates, non-residential graduate students, and student athletes) based on their observed case incidences. The testing proportions were determined by the modeling subgroup and the university COVID-19 taskforce.

Following universal testing of students prior to and within two weeks of arrival, we implemented surveillance testing based on model results. Beginning on the third week, 25% of non-residential undergraduates, 15% of graduate students and residential undergraduates, and 80% of student athletes were selected for testing. If a student tested positive, they were removed from the sampling frame for 90 days under the assumption that they had acquired natural immunity and based in Centers for Disease Control and Prevention guidance that did not recommend PCR testing within 90 days of confirmed infection [34]. In response to an increase in cases beginning in October, the sampling rate for non-residential students was adjusted to 35%. Following a super-spreader event in late October that caused a marked increase in cases during the first week of November, the University increased the proportion of students tested and implemented new mitigation policies, including switching to online course instruction and restricting all students to their dormitories for 14 days. In the second week of November, every student was tested. A limited number of in-person classes resumed the week before Thanksgiving. In line with most other IHEs, students remained home after the Thanksgiving Break and classes continued remotely.

### Observed incidence rate and main-scenarios comparison

Fig 1 provides the observed cumulative reported cases of COVID-19 at Quinnipiac University between September 16, 2020, and Nov. 24, 2020, as well as the predicted cumulative reported cases and predicted cumulative infections under our four main epidemiological scenarios. There was a total of 613 observed COVID-19 cases during this period (Fig 1). Cases were not evenly distributed across this period. Before October 11, there were only six reported cases. The infection rate gradually increased through October before an off-campus super-spreader event at the end of the month led to 495 cases between Nov. 1 and Nov. 6.

The fit between the observed case incidence and those predicted by the model also changed through time. Until Nov. 8, the $R_0 = 1.5$ scenarios fit the observed cumulative cases relatively well, with the 10 contacts/day scenario performing slightly better than the 8 contacts/day scenario (Fig 1). The observed cases fell within the $R_0 = 1.5$, 10 contacts/day scenario's prediction intervals on Sept. 6 and between October 25 and November 1. Although the observed cumulative cases between September 13 and October 18 were below this scenario's prediction intervals, the difference between the observed cases and the interval's lower boundary was no more than six cases between September 13 and September 20 and between October 11 and October 18. After Nov. 1, no scenario's prediction interval for diagnosed cases contained the observed cumulative cases. However, the observed cumulative cases fell within the $R_0 = 2.5$, 10 contacts/day scenario's prediction interval for total infections throughout this period.

Qualitatively, the model predicted steady increases in the case incidence under all four scenarios. The epidemiological trajectory in the $R_0 = 1.5$ scenarios were similar, regardless of the contact rate, whereas the contact rate differentiated the $R_0 = 2.5$ scenarios to a more noticeable degree. This same pattern occurred in each scenario's final median cumulative infections ($R_0 = 1.5$, 8 contacts/day: 178 infections, $R_0 = 1.5$, 10 contacts/day: 182 infections, $R_0 = 2.5$, 8 contacts/day: 438 infections, $R_0 = 2.5$, 10 contacts/day: 484 infections). In contrast, the observed

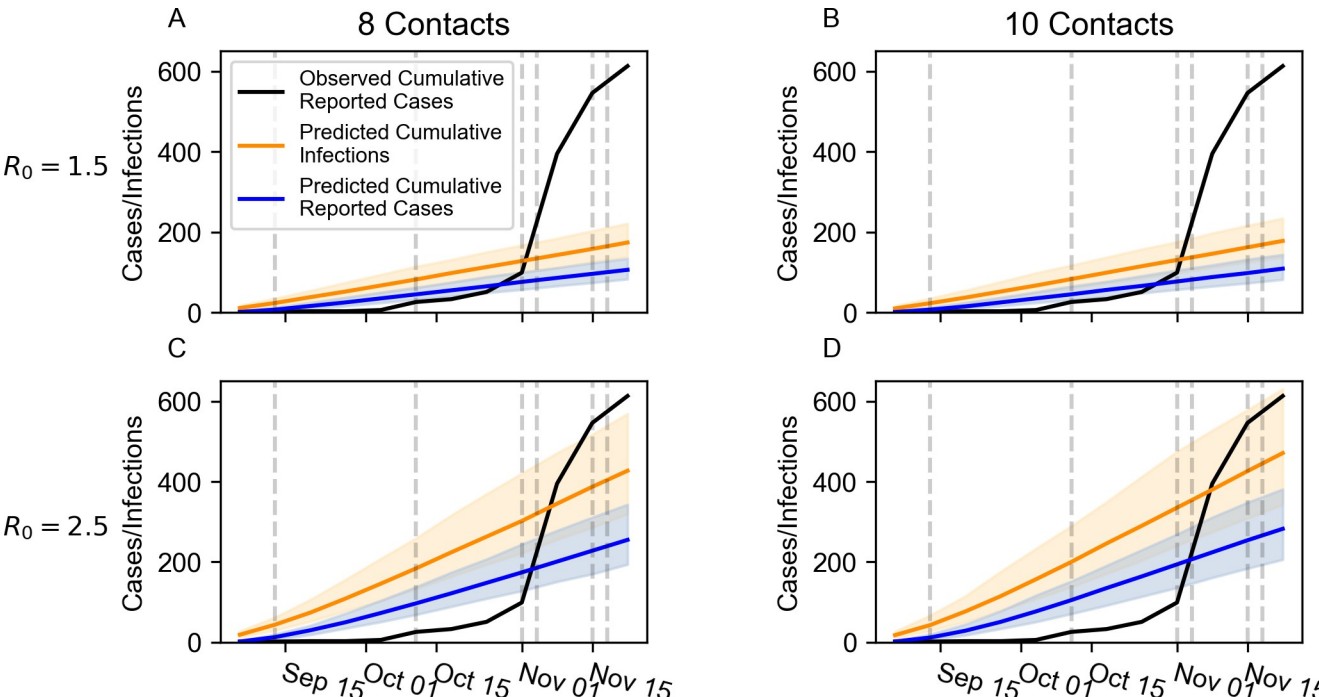

**Fig 1. Predicted cumulative infections and predicted cumulative cases under four scenarios compared to observed cumulative cases.** Solid, colored lines represent the median prediction across 1,000 replicate simulations. Shaded regions represent 90% prediction intervals spanning the 5th and 95th percentiles across the same 1,000 replicate simulations. Predicted cumulative infections are depicted in orange. Predicted cumulative reported cases are depicted in blue. Solid black lines depict observed cumulative reported cases. Vertical dashed lines indicate dates where the sampling scheme was adjusted. A) $R_0 = 1.5$, 8 contacts/day, B) $R_0 = 1.5$, 10 contacts/day, C) $R_0 = 2.5$, 8 contacts/day, D) $R_0 = 2.5$, 10 contacts/day.

cumulative cases increased at a heterogenous rate. Early in the semester, there were few observed cases. Beginning in October, the cumulative incidence began to rise at a greater, though still largely steady, rate. After the super-spreader event, the cumulative incidence sharply increased and grew nonlinearly.

## Sensitivity analysis for the rate of imported infections

To assess how the rate of imported infections impacts the model's epidemiological trajectory, we repeated all four scenarios but scaled the rate of imported infections by one half and two relative to the values in Table 2. As expected, the median total infections in each scenario was roughly proportional to the rate of imported infections (Fig 2). The proportional relationship is stronger when $R_0 = 1.5$ than when $R_0 = 2.5$. The estimated scaling constant for this relationship was similar between the $R_0 = 1.5$ scenarios ($R_0 = 1.5$, 8 contacts/day: 17.10; $R_0 = 1.5$, 10 contacts/day: 18.67). The estimated scaling constants for the $R_0 = 2.5$ scenarios were greater in magnitude than in the $R_0 = 1.5$ scenarios and were better differentiated between contact rates ($R_0 = 2.5$, 8 contacts/day: 24.12; $R_0 = 2.5$, 10 contacts/day: 27.63).

## Comparison of sampling strategies

Predictions for final cumulative infections differed minorly to moderately across targeted testing, SRS, and complete sampling when $R_0 = 1.5$ (Fig 3A and 3B). The median predictions for targeted testing and SRS differed by no more than three infections, whereas the difference between complete sampling and targeted sampling was 19 infections under 8 contacts/day and 24 infections under 10 contacts/day. The lower boundaries on the 90%

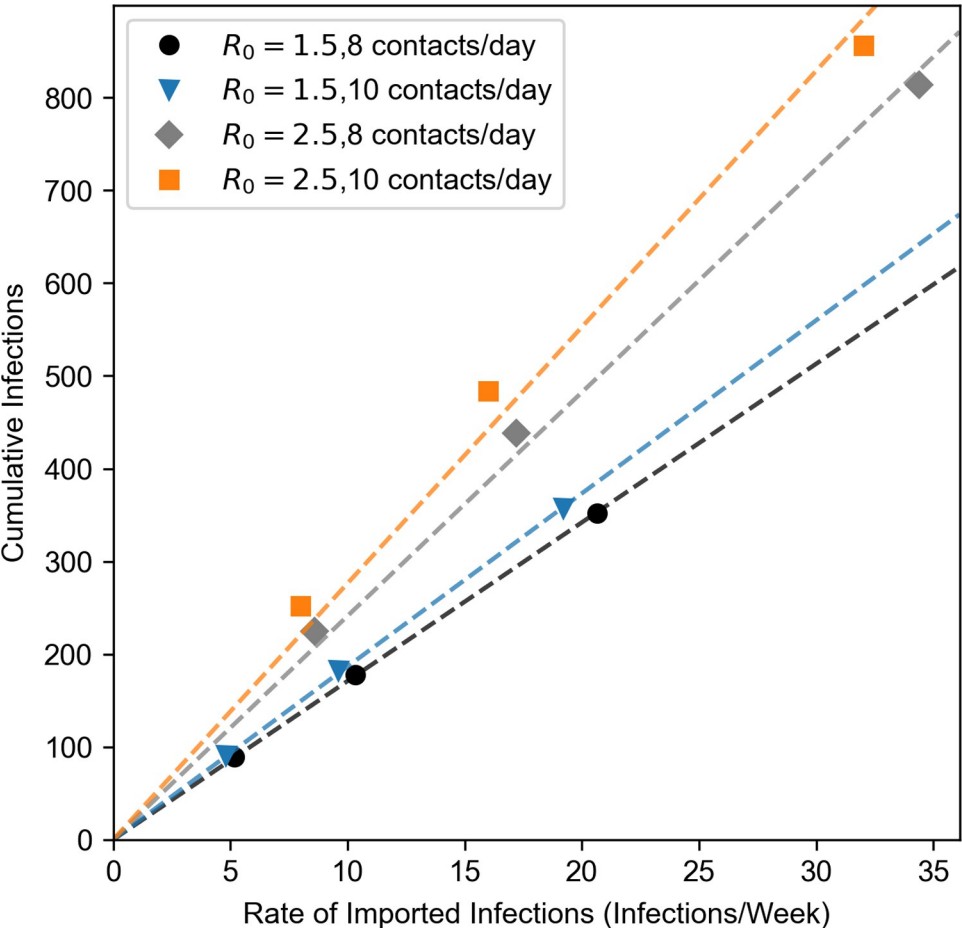

**Fig 2. Median cumulative infections against rate of imported infections under four epidemiological scenarios.**
Dashed lines represent the least-squares line for its corresponding scenario, constrained to pass through the origin.
Black circles: $R_0$ = 1.5, 8 contacts/day, blue triangles: $R_0$ = 1.5, 10 contacts/day, grey diamonds: $R_0$ = 2.5, 8 contacts/day,
orange squares: $R_0$ = 2.5, 10 contacts/day.

prediction intervals (*i.e.*, 5th quantile) differed less. No more than 16 infections separated the lower boundary of any design under either scenario. The differences between complete sampling and the random sampling strategies were more pronounced in the 90% predictions intervals' upper boundaries (*i.e.*, 95th quantile). Relative to targeted sampling, complete sampling had 32.05 fewer infections at the upper boundary under 8 contacts/day and 44 fewer infections under 10 contacts/day. In contrast, targeted sampling and SRS had very similar upper boundaries that differed by no more than three infections.

When $R_0$ = 2.5, complete sampling led to many fewer infections relative to the random sampling strategies (Fig 3C and 3D). In the $R_0$ = 2.5, 8 contacts/day scenario, the median cumulative infections was 110 infections lower under complete sampling relative to targeted sampling and was 146 infections lower in the $R_0$ = 2.5, 10 contacts/day scenario. Once again, the upper boundaries on the 90% confidence intervals showed even more pronounced differences between complete sampling and targeted sampling (8 contacts/day: 182 infections, 10 contacts/day: 206.85 infections). The differences between targeted sampling and SRS, however, were comparably minor. The median predictions differed by no more than two infections and the lower boundaries on the prediction intervals differed by no more than 9.05 infections. The

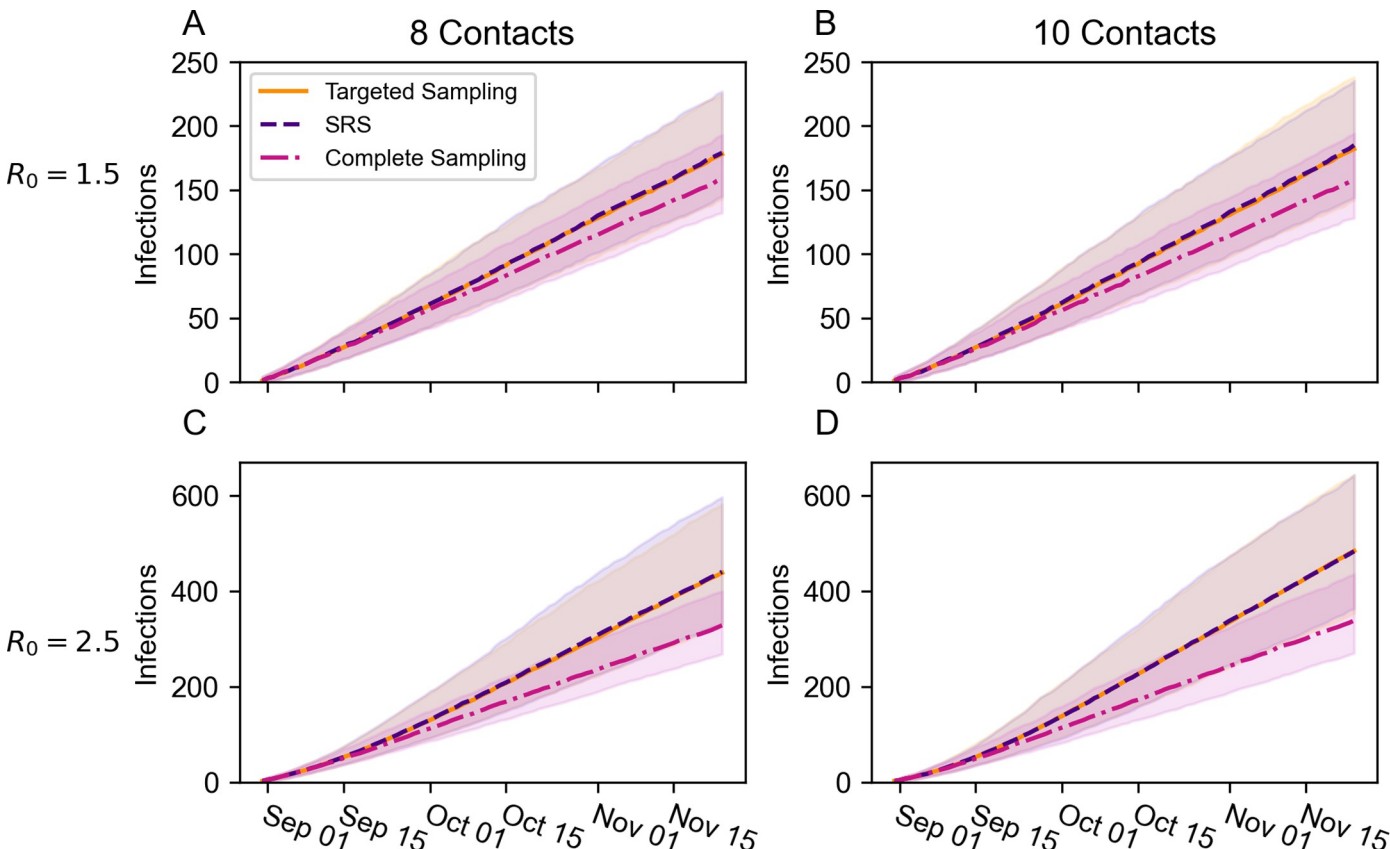

**Fig 3. Comparison of sampling designs for surveillance testing based on predicted cumulative infections under four scenarios.** Solid and dashed lines represent the median predicted cumulative infections across 1,000 replicate simulations. Shaded regions represent 90% prediction intervals spanning the 5th and 95th percentiles across the same 1,000 replicate simulations. Targeted sampling is depicted in orange. Simple random sampling (SRS) is depicted in purple. Complete sampling is depicted in magenta. A) $R_0 = 1.5$, 8 contacts/day, B) $R_0 = 1.5$, 10 contacts/day, C) $R_0 = 2.5$, 8 contacts/day, D) $R_0 = 2.5$, 10 contacts/day.

greatest difference between these two scenarios occurred in the upper boundary under 8 contacts/day (15 fewer infections under targeted testing). The difference was minute, however, when there were 10 contacts/day (one fewer infection under targeted testing).

Under most holdout scenarios, the median cumulative infections and lower boundaries on the 90% prediction intervals differed very little between targeted sampling and SRS (Fig 4). On the last day of the simulation, the medians and the lower boundaries for targeted sampling and SRS were all within three cases of each other under the all-interventions, no-contact-tracing, and delayed-symptomatic-testing scenarios. The no-symptomatic-testing scenario was the exception. In this case, the median prediction for SRS was 19 infections greater than that for targeted sampling, and the lower boundary on SRS's prediction interval was 83.3 infections greater than that of targeted sampling. In contrast, targeted sampling had 90% confidence intervals with noticeably lower upper boundaries than SRS under every holdout scenario (all-interventions: 14.0 infections, no-contact-tracing: 19.2 infections, delayed-symptomatic-testing: 18.0 infections, no-symptom-monitoring: 38.8 infections). The difference between the two strategies' upper boundaries displayed an upward global trend through time. This may indicate that the difference between the two strategies would have become even more pronounced if the simulations were allowed to continue, although each scenario had a declining local trend on the simulation's last day. It is also worth noting that there was substantial variation in how many infections were predicted in each scenario. Under targeted sampling, the all-

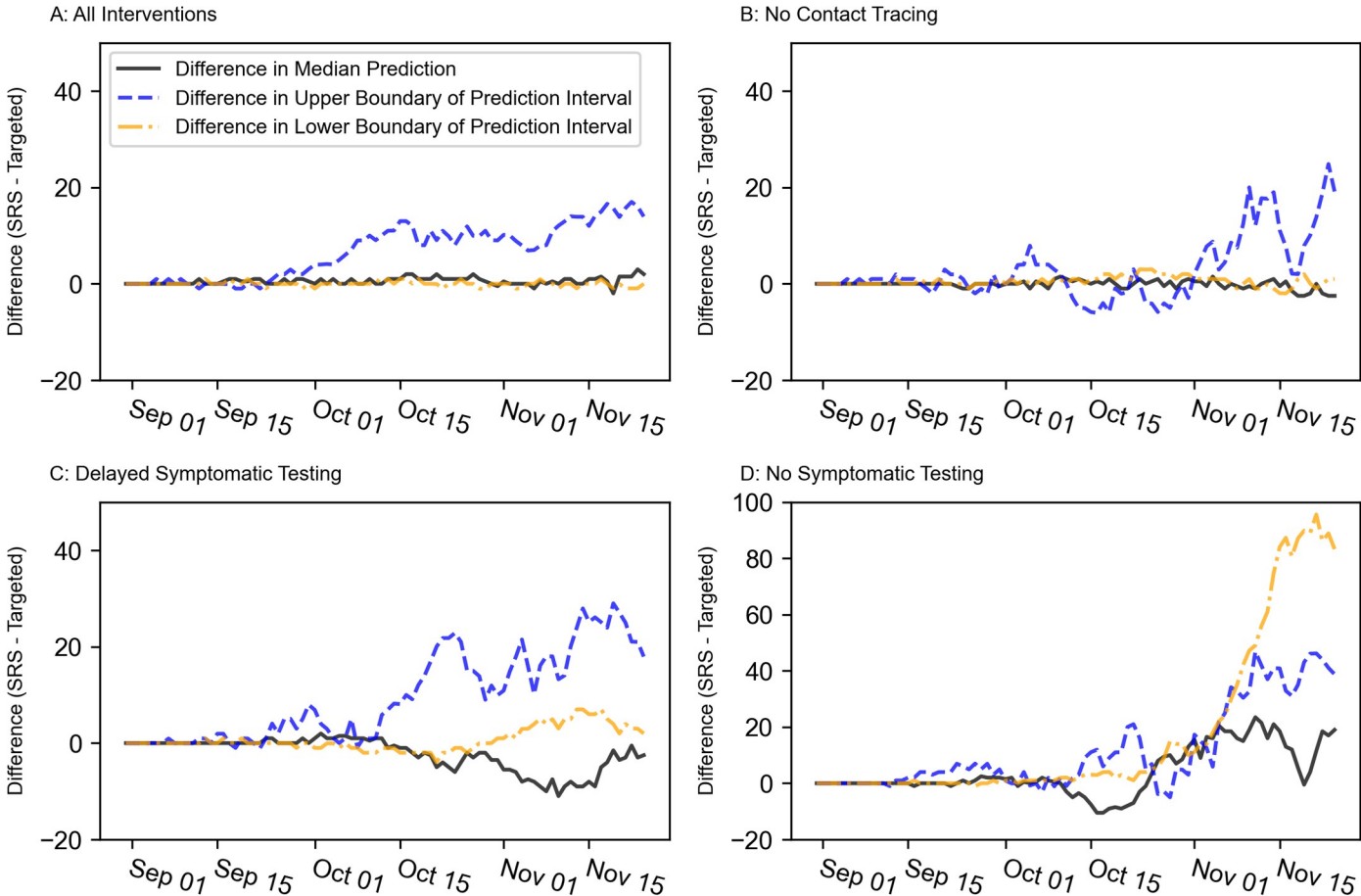

**Fig 4. Differences between the cumulative infections under targeted sampling and SRS when other interventions are removed.** Black, solid lines represent the difference between the median predicted cumulative infections under SRS and that under targeted sampling. Blue, dashed lines represent the difference between the 90% prediction interval's upper boundary under SRS and that under targeted sampling. Gold, dashed-and-dotted lines represent the difference between the 90% prediction interval's lower boundary under SRS and that under targeted sampling. A) All-interventions scenario, B) No-contact-tracing scenario, C) Delayed-symptomatic testing scenario, D) No-symptomatic-testing scenario.

intervention scenario had 139 infections, the no-contact-tracing scenario had 212.5 infections, the delayed-symptomatic-testing scenario had 318.5 infections, and the no-symptomatic test-ing scenario had 1,458 infections.

## Discussion

### Overview of university response

Like most IHEs, Quinnipiac University developed a plan to provide an in-person learning experience in fall 2020 while mitigating the spread of SARS-CoV-2. Given the novelty of COVID-19, there was limited data available to guide policy decisions. Therefore, the University extended an existing model of SARS-CoV-2 dynamics to help inform the University's COVID-19 policies. The modeling team devised, implemented, and updated a targeted sampling scheme to choose students for surveillance testing with the goal of quickly detecting and then responding to outbreaks within university residences. Finally, the University had a devoted team of contact tracers and asked students to actively monitor themselves for symptoms using a cellular phone application.

Many research groups housed across several universities used models of COVID-19 dynamics to inform university re-opening plans for fall 2020. These models utilized varied strategies, including not only agent-based models [14, 35–38], but also models based on classic compartmental structures (e.g., [15, 28, 39–44]), network theory [22, 45], and probability theory [46]. Our agent-based model was a specially modified version of the Institute for Disease Modeling's Covasim model [16]. Extending Covasim, rather than building our own model from scratch, allowed us to capitalize on a well-structured model of disease dynamics that incorporated the best estimates for epidemiological parameters that were available in summer 2020 [16]. Our limited personnel could then focus on refining the model to account for social and epidemiological properties unique to our situation, such as the structure of university-owned residences. Because the University's response to COVID-19 was so time sensitive, it would not have been feasible to develop such a complicated model without drawing on publicly available, open-source programs, and such programs could likely facilitate university responses to future EIDs and Disease X scenarios.

Even with the intellectual head start that the original Covasim provided us, certain policy decisions had to be made using versions of the model that lacked features described here. For instance, decisions about what sampling coverage to use for surveillance testing and whether to use our targeted sampling design rather than SRS had to be informed using a version of the model that only simulated residential students, did not incorporate dedensification in university-owned properties, and set the probability of transmission to Covasim's default value. Indeed, policy decisions made for spring 2020 and fall 2021 were informed using model features that were not described here because they were introduced in newer versions of Covasim (*e.g.*, waning immunity, vaccinations, and co-circulating strains [47]) or were created after fall 2020 (*e.g.*, a separate graduate student category for agents).

## Model performance

One of the primary roles that our model played in Quinnipiac's COVID-19 response was to assess how disease properties that were not well understood in summer 2020 impacted case and infection rates. In the work described here, we explored four main scenarios that differed in basic reproductive number ($R_0$) and average contact rate: i) $R_0$ = 1.5, 8 contacts/day, ii) $R_0$ = 1.5, 10 contacts/day, iii) $R_0$ = 2.5, 8 contacts/day, iv) $R_0$ = 2.5, 10 contacts/day. No scenario universally fit the observed case incidence data well. The $R_0$ = 1.5, 10 contacts/day scenario performed adequately until a super-spreader event led to a sudden shock in the observed incidence rate on Nov. 8 (Fig 1B). The observed cumulative cases on and after Nov. 8 exceeded those predicted in every scenario. However, the University substantially increased its surveillance testing efforts during this period in response to the super-spreader event and moved to sampling all students on Nov. 15 (Table B in S1 Appendix). Since no scenario included this shift in effort, we would expect the observed case rate to exceed that of the model, even if the disease dynamics were accurately simulated. However, the predicted cumulative infections provided an absolute cap on how many diagnosed cases could have occurred in the model. Since the observed cumulative cases on and after Nov. 8 remained within the prediction interval for cumulative infections under the $R_0$ = 2.5, 10 contacts/day scenario, the model may have still provided a reasonable approximation to disease dynamics during this period (Fig 1D). Differences between the model's sampling rate and the achieved sampling rate during testing may have also contributed to the model's tendency to overpredict the number of reported cases early in the semester. Although the model's sampling rates and the requested sampling rate during testing initially matched, frequently only around 70% of students selected for surveillance testing actually responded.

It is noteworthy that both scenarios that fit the observed data best included an average of 10 contacts/day. Model-based estimates for the student contact rate at Villanova University were also around 10 contacts/day ([8.2020,10.9953]) [44], suggesting that this may be a reasonable approximation to university student behavior under pandemic conditions. However, further estimates from a more diverse set of IHEs are needed.

The scenarios' shifting fits indicate that the model may have performed better if model parameter values were dynamic. As previously suggested, the $R_0 = 2.5$, 10 contacts/day scenario's 90% prediction interval for cumulative diagnosed cases could have bounded the observed cases if the surveillance-testing interventions' sampling rates had shifted within the simulation. Shifts in other model parameters could have allowed a single scenario to accurately represent the University's disease dynamics and to better capture the curvature in the observed cases' trajectory. Periods where parameter shifts would improve the fit may reflect changes in the University's disease dynamics. The two best fitting scenarios used the same contact rate but differed in $R_0$, which under the model's assumptions, suggests a shift in the probability of transmission. Such a shift would occur, for instance, if students became lax toward public health policies as the semester progressed (*e.g.*, violating the universal mask policy more frequently). In addition, the $R_0 = 2.5$ scenario had a higher rate of imported infections, which may reflect the rising COVID-19 prevalence within the surrounding community as the semester progressed.

Alternatively, the model's fit may have also been improved by representing the super-spreader event as a sudden increase in the number of infections without any in-simulation transmission. This strategy would have emphasized the uniqueness of the super-spreader event as a violation of the University's usual disease dynamics. A compartmental model of COVID-19 transmission at Villanova University provided a better fit to observed case numbers when it was allowed to include similar shocks [44].

## Relationship between IHEs and their surrounding communities

Both the observed cumulative cases and the model results underscore the interconnection between student populations and an IHE's surrounding community. The rate of imported infections is the model's primary representation of how the student body interacts with the surrounding community. The model's assumptions favor a proportional relationship between the final cumulative cases and the rate of imported infections. We therefore measured the model's sensitivity to the rate of imported infections using the slope of a least-squares line passing through the origin that relates the cumulative number of infections to the rate of imported infections. If the rate of imported infections had no net effect on local transmission, we would expect the line's slope to be 12.29 because there are 12.29 weeks (*i.e.*, 86 days) in each simulation. However, the slope exceeded this limit in every scenario. The model becomes more sensitive to the rate of imported infections at higher $R_0$ values and higher contact rates. In the $R_0 = 1.5$, 8 contacts/day scenario, one additional imported infection per week is estimated to cause 17.1 more infections by the end of the simulation, but the same increase is estimated to cause 27.6 more infections under the $R_0 = 2.5$, 10 contacts/day scenario. This latter slope is over twice the baseline rate of 12.3 infections, suggesting a significant role for on-campus-off-campus interactions in an IHE's disease dynamics. These findings corroborate results found in other models of IHE COVID-19 dynamics that suggested that imported infections can have a major effect on an IHE's incidence rate or can hinder the effectiveness of an IHE's interventions [14, 39, 46] (though see [35]). Interestingly, Gressman and Peck [36] found nearly the same proportional relationship between the cumulative number of infections and the daily contact rate in their model as we found between the cumulative number of infections

and the rate of imported infections under the $R_0 = 2.5$, 10 contacts/day scenario. The significance, if any, of this similarity is unclear, though.

Two key patterns in the observed cumulative cases suggest a linkage between campus disease dynamics and events in the surrounding community. First, after a low incidence rate in August and September, the University's rate considerably increased at the beginning of October (Fig 1). During this same interval, the case incidence in New Haven County, in which Quinnipiac is located, rose from an average of 26.73 new cases per day in September to 92.74 new cases per day in October [48], suggesting a possible relationship between community prevalence and campus prevalence. Although this study is not equipped to evaluate the nature of this proposed relationship, the roughly 7,100 students attending Quinnipiac in person during the fall 2020 semester is small relative to the 864,835 residents of New Haven County [11, 49]. Further, the case incidence in New Haven County had already begun to rise in September from an average of 20.45 cases per day in August [48]. As a result, it seems likely that increases in the surrounding community's prevalence led to increases in the University's incidence rate and not vice-versa. This interpretation would be consistent with the rate of imported infection's sensitivity analysis since the estimated prevalence in the community partially determined this rate (S1 Appendix).

The second major pattern in the observed case data was the incidence's sudden increase after the late October super-spreader event (Fig 1). This event was associated with an off-campus Halloween party. Although student behavior was the core cause of the event, the event was also linked to alleged violations of state public health laws at a business in the surrounding community [50]. So, despite being a campus-specific phenomenon, it arose from a combination of on-campus factors (*i.e.*, student behavior) and off-campus factors (*i.e.*, legal lapses in the community).

Here, we have focused on the role that the surrounding community may have played in increasing an IHE's case incidence rate, but both model-based and empirical evidence suggest that disease dynamics at IHEs can also increase the case and mortality rates in the surrounding community [51–54]. Further research is needed to better understand this complex relationship, including under what conditions IHEs have a negative impact on surrounding communities and under what conditions surrounding communities have negative impacts on IHEs. Currently, compartmental model results suggest that surrounding communities have a larger role in augmenting incidence rates at IHEs when the IHE has effectively controlled transmission through public health interventions [39].

The interconnection between campuses and communities may challenge containment efforts in the face of future EIDs. Ultimately, IHE officials only have jurisdiction over the campus itself. As such, successful containment may require close collaboration between university and local, county, and state officials [51, 55]. Yet, the interconnection also means that information about the surrounding community can inform on-campus policies. For example, a university could increase surveillance testing or impose more stringent requirements to enter campus when community transmission is high.

## Sampling strategy for surveillance testing

Like many IHEs, Quinnipiac conducted surveillance testing on the student body to detect asymptomatic and subclinical infections. Unlike most IHEs, residential students were sampled for surveillance testing using a targeted design that combined features of stratified and clustered sampling. The design's goal was to ensure even and consistent coverage across rooms, suites, and floors in university-owned properties, but the targeted design was partially extended to include externally owned properties if three or more non-residential students

were known to share the address. Observed cases in university-owned properties did tend to cluster within rooms and floors, suggesting that our strategy likely did facilitate the discovery of case clusters. Further, we divided the student body into residential students, non-residential undergraduates, non-residential graduate students, and student athletes. Each subpopulation received its own sampling rate. Because we had categorized students in this manner, we were able to rapidly increase testing in subpopulations where we had identified increased case rates. We were also able to determine whether any additional strategies employed to interrupt transmission in those subpopulations were effective.

To the best of our knowledge, no other IHE used the same sampling strategy as Quinnipiac. However, other universities had favorable results with similar targeted strategies. An Indiana university used a stratified sampling design and adapted each stratum's sampling rate in response to new case data, which likely contributed to the containment of a major outbreak [12]. Similarly, empirical and model-based evidence suggest that a targeted sampling design at Clemson University that focused surveillance tests towards university-own residences that experienced a new case played an important role in mitigating disease prevalence [15].

Simulation results suggest that our targeted sampling strategy and SRS would have had comparable effects on the infection rate when contact tracing and app-based symptom monitoring were also implemented (Fig 3). Complete sampling decreased the cumulative infections under every scenario, but the degree of difference relative to targeted sampling depended on $R_0$. In the $R_0 = 1.5$ scenarios, complete sampling reduced the median cumulative infections by no more than 24 infections, whereas the median cumulative infections was reduced by at least 110 infections in both $R_0 = 2.5$ scenarios. Assuming comparable results would occur using the effective reproductive number, these results validate the University's strategy of shifting sampling rates based on current incidence data. If a university has limited resources to test for an EID, a more economical strategy for surveillance testing may be to sample the student population at a lower rate when the observed transmission rate is low because the reduction in infections per test will also be lower. This would then reserve resources for increased sampling during periods of high transmission when the reduction in infections per test is high. A similar strategy appeared to provide favorable results at a college in South Carolina [13]. This conclusion is similar to Paltiel *et al.* [28], who found that the effective reproductive number altered what sampling frequency was most cost effective, and to Hambridge, Kahn, and Onnela [38], who found that increasing the frequency of surveillance testing has smaller effects on the infection rate when $R_0$ is low than when it is high. A university using this adaptive sampling design would need to implement other public health interventions, though. The model results were predicated on contact tracing and app-based symptom monitoring's inclusion, and surveillance testing in the low-rate phase would predominantly monitor whether the rate needs to be shifted upward, so other inventions would be necessary to regulate transmission.

In the holdout scenarios, we withheld individual interventions in the model to explore each intervention's impact on a population of residential students. These results suggest that targeted sampling and SRS had comparable effects on the infection rate in our main epidemiological scenarios because other public health interventions were effectively controlling transmission. When either contact tracing or symptom monitoring was absent, the targeted strategy provided some benefit over SRS (Fig 4B–4D). Interestingly, under most holdout scenarios, the major difference between strategies was in the upper boundary of the 90% prediction intervals rather than the median prediction. Given that complete sampling also had a greater impact on the upper boundary, the sampling strategy used for surveillance testing may have more value in mitigating the worst-case outcome than the typical outcome. Previous work has similarly suggested that the frequency of surveillance testing may have a greater role in controlling the maximum size of outbreaks than the average size [14]. Which intervention is

removed also influences targeted sampling's effect. In the no-contact-tracing and delayed-symptom-monitoring scenarios, targeted sampling had a noticeable but modest impact on the 90% prediction interval's upper boundary and no noticeable impact on the lower boundary or median prediction. In contrast, targeted sampling caused sizeable reductions in both the prediction interval's boundaries and even moderately reduced the median prediction.

## App-based symptom monitoring as a containment strategy

Comparisons between the holdout scenarios under targeted sampling suggest that app-based symptom monitoring may have a high efficacy for controlling transmission. App-based symptom monitoring was assumed to reduce the expected waiting time between when a student developed symptoms and when the student sought testing and quarantined. We set this expected waiting time to one day in the main and all-intervention scenarios. The delayed-symptom-monitoring scenario assumed the symptom-monitoring application was missing, and the students' expected waiting times for testing were 4.82 days, the same as the general population [33]. The median prediction for final cumulative infections under the all-interventions scenario had 179.5 fewer infections than that of the delayed-testing scenario. This difference is not only twice as great as that of the no-contact-tracing scenario but is even greater than the difference between targeted sampling and complete sampling under the $R_0 = 2.5$, 10 contacts/day scenario. App-based monitoring's apparent efficacy in the model results may arise because the symptom-monitoring application instructed symptomatic students to immediately quarantine after seeking testing but before receiving results; in contrast, surveillance testing and contact tracing both delayed quarantining until test results became available. Other models have found similar relationships between testing delays and surveillance testing's efficacy, either through increased effectiveness when testing is implemented more frequently [28, 39, 42] or when test results are returned more quickly [35, 39, 42, 46]. However, the actual impact that app-based monitoring had on the University's COVID-19 dynamics is less clear. Our model assumed every student faithfully used the application daily and would report symptoms to the app earlier than they would otherwise seek treatment. In reality, community uptake was limited, and no data is available on whether the application altered student behavior.

## Role of non-residential students in disease transmission

Finally, since the $R_0 = 2.5$, 10 contacts/day and all-interventions scenarios were identical except for the inclusion of non-residential students in the former, a comparison between the two provides insights into the role that non-residential students may have played in disease transmission. Even though residential students comprised 49.0% of the student population in the $R_0 = 2.5$, 10 contacts/day scenario, the cumulative infections under the all-interventions scenario was 38.9% that of the $R_0 = 2.5$, 10 contacts/day scenario after subtracting the expected number of imported infections from each scenario's cumulative infections. This suggests that non-residential students have a disproportional impact on a university's local transmission. Indeed, the University's observed testing data supported this hypothesis. The gradual case increases that began in the first week of October were due to an increase in cases in non-residential students. Other universities also witnessed higher case rates in non-residential students than their residential peers, although the relative sizes of these subpopulations were not always clear [9, 12]. In these scenarios, non-residential students may have experienced high infection rates through a combination of increased mixing with the surrounding community and decreased oversight from the University's public health team. However, this pattern was far from universal, and other universities reported a disproportional number of cases in their residential populations, although the difference was not always statistically significant [10, 13, 15].

## Implication for future emerging infectious diseases

The implications these results have for future EID outbreaks is complicated. App-based symptom monitoring heavily reduced the infection rate in the model and requires few resources to implement. Further, because symptom monitoring is a strategy that translates to most diseases, app-based symptom monitoring would be relatively easy to launch in the face of a new Disease X scenario. Yet, individual students are responsible for monitoring their own health under this intervention, and as the late October super-spreader event and comparable events at other universities (*e.g.*, [12, 13, 44]) illustrate, student behavior can be difficult to predict. A website-based symptom monitoring program failed to prevent a major outbreak at an Indiana university [12]. University officials may have more success with app-based monitoring if they include measures to encourage adherence. Indeed, the Indiana university attributed their successful containment efforts partially to a public health educational campaign [12], and model-based results indicate that educational programs can decrease infection rates when combined with other interventions [42]. In any case, app-based symptom monitoring would need to be used in conjunction with other interventions in a Disease X scenario, as its effectiveness will depend on disease-specific rates and severities of symptoms.

Contact tracing was also effective in controlling the transmission rate in simulated results. This intervention has a long and proven track record in public health policy, translates well to novel diseases, and was found to have a comparable effectiveness to surveillance testing with complete or nearly complete sampling in other modeling studies of COVID-19, at least under certain measures [44]. However, contact tracing requires trained personnel that may not be available at every university.

Finally, like numerous other modeling studies (*e.g.*, [28, 44]), we found that surveillance testing with complete sampling can substantially decrease infection rates. Complete sampling requires a vast amount of financial, logistical, and human resources to implement, though, and cannot be launched in Disease X situations until reliable tests are developed. Many universities will not have the means to make this strategy feasible. Even when complete sampling is infeasible, though, targeted sampling and SRS can still assist in outbreak detection at a significantly reduced cost, as it did for Quinnipiac. Although the model results found that the use of targeted sampling to obtain students for surveillance testing did not impact the infection rate differently than SRS, there are still reasons why a university might choose to use targeted sampling in the face of an EID or Disease X scenario. First, the holdout scenarios suggested that targeted sampling does control worst-case infection rates better than SRS when the delay between developing symptoms and testing is increased. This suggests that targeted testing's relative advantage depends on features of the disease's presentation, such as the severity of illness, the rate of asymptomatic infections, and degree of infectiousness before symptom onset. These properties are variable between diseases and may not be well-understood early in an EID outbreak. Targeted testing does not require many more resources than SRS, so a university might elect to use targeted sampling to capitalize on any potential benefits with little penalty. Second, the even sampling coverage that targeted testing provides can help to reassure students, faculty, staff and parents that a campus is safely opening. Although not directly related to disease progression, the reduction in anxiety and stress could yield significant mental health benefits.

Disease X scenarios pose major threats to global public health because policy responses need to be made before a substantial amount of information is available about the disease. COVID-19 was one example of such a scenario, so careful documentation and study of how institutions, such as IHEs, responded to COVID-19 could improve responses to future Disease X scenarios. Here, we described Quinnipiac University's response and its effectiveness.

However, IHEs contain diverse populations and exist in varied contexts that can influence an intervention's effect. For example, Quinnipiac University and a college in South Carolina launched comparable COVID-19 responses (*e.g.*, app-based symptom monitoring, adaptive sampling rates for surveillance testing), but the case data for the two universities sometimes yielded contradictory patterns (*e.g.*, the proportion of off-campus students in the case data) [13]. Thus, one major limitation to this study is the limited scope of its population. A more thorough review of IHE responses and their effectiveness would help to build more robust responses. Although some attempts have been made to create such a review [52], limited and inconsistent documentation across IHEs have stifled these efforts. A second major limitation is that COVID-19 is just one Disease X scenario. The next Disease X may have very different medical and epidemiological properties to COVID-19 and strategies effective for COVID-19 may not be effective in these scenarios. Public health and IHE officials will therefore need to remain flexible when preparing for future EIDs and Disease X scenarios and build contingency plans into their responses to account for differential effectiveness.

## Supporting information

**S1 Appendix. Derivation of model parameters and detailed targeted sampling description.** (DOCX)

**S1 File. Python and R scripts to replicate the analyses in this paper.** (ZIP)

## Acknowledgments

The authors would like to thank the Quinnipiac University COVID-19 Task Force for supporting the modeling efforts, to the faculty and staff for their assistance in data collection, to N Bharti for providing us with a preprint of her and her collaborator's manuscript on university-based contact tracing data, as well as M Ferrari for introducing us to N Bharti, and CC Kerr and the Institute for Disease Modeling for access to the Covasim model.

## Author Contributions

**Conceptualization:** K. James Soda, Xi Chen, Richard Feinn, David R. Hill.

**Formal analysis:** K. James Soda.

**Methodology:** K. James Soda, Xi Chen, Richard Feinn, David R. Hill.

**Project administration:** Richard Feinn, David R. Hill.

**Software:** K. James Soda.

**Visualization:** K. James Soda, Xi Chen.

**Writing – original draft:** K. James Soda, Xi Chen, Richard Feinn, David R. Hill.

**Writing – review & editing:** K. James Soda, Xi Chen, Richard Feinn, David R. Hill.

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
