## [Decision Letter · Decision Letter 0]

12 Mar 2023

PONE-D-23-00956

Monitoring and responding to emerging infectious diseases in a university setting: A case study using COVID-19

PLOS ONE

Dear Dr. Soda,

Thank you for submitting your manuscript to PLOS ONE. After careful consideration, we feel that it has merit but does not fully meet PLOS ONE’s publication criteria as it currently stands. Therefore, we invite you to submit a revised version of the manuscript that addresses the points raised during the review process.

We look forward to receiving your revised manuscript.

Kind regards,

Barbara T Rumain, PhD

Academic Editor

PLOS ONE

Journal Requirements:

Reviewers' comments:

Reviewer's Responses to Questions

**Comments to the Author**

1. Is the manuscript technically sound, and do the data support the conclusions?

Reviewer #1: Yes

Reviewer #2: Yes

2. Has the statistical analysis been performed appropriately and rigorously? 

Reviewer #1: Yes

Reviewer #2: Yes

3. Have the authors made all data underlying the findings in their manuscript fully available?

Reviewer #1: Yes

Reviewer #2: Yes

4. Is the manuscript presented in an intelligible fashion and written in standard English?

Reviewer #1: Yes

Reviewer #2: Yes

5. Review Comments to the Author

Reviewer #1: The authors present a modified agent-based covid-19 model based on Covasim. The modifications are explained in detail and are based on disease mitigation strategies implemented by Quinnipiac University in the Fall 2020 semester. The results of the model are then compared to observed data from that semester. Suggestions for future infectious diseases at institutes of higher educations are also provided.

The model and methodology for analyzing it appear sound and thorough. My main questions are regarding the main four scenarios:

Can the authors provide the rationale for choosing the two specific R_0 values used in the study? Did these come from known values at the time or were they completely hypothetical?

Also, the authors cite Bharti et al. for the average contact rates of 8 and 10, which was published in 2021. However, the authors state that their model was created in summer 2020 to help plan for fall 2020. Was the preprint mentioned in the Acknowledgments available to the authors for their initial use of the model? This point could be made clear.

Reviewer #2: This manuscript provides insight into one university's experience with COVID-19 modeling and disease mitigation. It provides detailed information related to disease modeling strategies and observed disease transmission that would be useful to other IHEs for future use of Disease X modeling. Additionally, they present practical considerations related to university prevention strategies. Comparisons with other IHE experiences highlight similar strategies/outcomes as well as differing experiences that call for further investigation.

To provide additional clarity for the reader in understanding the figures, the following edits are recommended:

Given that Figure 1 includes predicted and observed values, include "predicted" in the manuscript description and the figure labeling. With regard to Figure 3, the manuscript text does state that the figure shows predicted values; adding this to the figure label would provide further clarity. Label y-axis for figures 1 and 3.

6. PLOS authors have the option to publish the peer review history of their article (what does this mean?). If published, this will include your full peer review and any attached files.

**Do you want your identity to be public for this peer review?** For information about this choice, including consent withdrawal, please see our Privacy Policy.

Reviewer #1: No

Reviewer #2: No

---

## [Author Response · Author response to Decision Letter 0]

12 Apr 2023

Responses to Reviewer #1

Comment: The authors present a modified agent-based covid-19 model based on Covasim. The modifications are explained in detail and are based on disease mitigation strategies implemented by Quinnipiac University in the Fall 2020 semester. The results of the model are then compared to observed data from that semester. Suggestions for future infectious diseases at institutes of higher educations are also provided.

Response: We believe that this is an accurate summary of our work and goals. 

Comment: The model and methodology for analyzing it appear sound and thorough.

Response: We thank the reviewer for this favorable assessment. 

Comment: My main questions are regarding the main four scenarios:

Can the authors provide the rationale for choosing the two specific R_0 values used in the study? Did these come from known values at the time or were they completely hypothetical?

Response: Our main four scenarios used two values for R0, 1.5 and 2.5. Both these values were informed and later validated by the literature. We selected the R0 value of 2.5 because the (US) Centers for Disease Control and Prevention’s COVID-19 Pandemic Planning Scenarios, as published on May 20, 2020, listed 2.5 as their best estimate for R0. In addition, Li et al. (2020) estimated a similar value for R0 (2.2) using case data in Wuhan, China from between December 10, 2019, and January 4, 2020. 

Our model did not directly use R0 as a model parameter. Instead, we used R0 to establish reasonable values for the probability that a social contact between an infectious individual and a susceptible individual would lead to disease transmission. However, we also believed that public health interventions, such as social distancing and masking, would also lower the probability of transmission. To account for these effects, we also incorporated scenarios where R0 was 1.5, partially in convention with one of the effective reproductive numbers used in Paltiel, Zheng, and Walensky (2020). Later, Pitzer et al.(2021) retroactively validated this choice through their estimates for the effective reproductive rate in the United States during the summer of 2020.

The rationale provided above, along with the relevant citations, have now been added to the manuscript. (Relevant Page and Line in Manuscript with Tracked Changes: pg. 11, Lines 230-238)

Comment: Also, the authors cite Bharti et al. for the average contact rates of 8 and 10, which was published in 2021. However, the authors state that their model was created in summer 2020 to help plan for fall 2020. Was the preprint mentioned in the Acknowledgments available to the authors for their initial use of the model? This point could be made clear.

Response: The reviewer is correct that Dr. Bharti provided us with a preprint of her and her colleagues’ paper before it was published, and this preprint informed our work. We now explicitly state this in the main text. (Relevant Page and Line in Manuscript with Tracked Changes: pg. 11, Lines 238-240)

Response to Reviewer 2

Comment: This manuscript provides insight into one university's experience with COVID-19 modeling and disease mitigation. It provides detailed information related to disease modeling strategies and observed disease transmission that would be useful to other IHEs for future use of Disease X modeling. Additionally, they present practical considerations related to university prevention strategies. Comparisons with other IHE experiences highlight similar strategies/outcomes as well as differing experiences that call for further investigation.

Response: We believe that this is an accurate summary of our work. 

Comment: To provide additional clarity for the reader in understanding the figures, the following edits are recommended:

Given that Figure 1 includes predicted and observed values, include "predicted" in the manuscript description and the figure labeling.

Response: We now explicitly mention that Figure 1 contains both predicted and observed values, both within the in-text description and in the figure title, and are more careful to differentiate predicted results from observed results. The figure legend has also been updated to emphasize that the median predictions and confidence intervals were based on the same 1,000 replicate simulations. (Relevant Page and Line in Manuscript with Tracked Changes: pg. 16, Lines 343-347; pg. 16, Lines 351-352; pg. 16, Line 354)

Comment: With regard to Figure 3, the manuscript text does state that the figure shows predicted values; adding this to the figure label would provide further clarity.

Response: Both the figure title and figure legend now explicitly state that Figure 3 shows predicted values. Just as in Figure 1, we also updated the legend to emphasize that the median predictions and confidence intervals were based on the same 1,000 replicate simulations. (Relevant Page and Line in Manuscript with Tracked Changes: pg. 19, Lines 407-408; pg. 19, Lines 410)

Comment: Label y-axis for figures 1 and 3.

Response: Figures 1 and 3 now have labelled y-axes. (Relevant Files: Fig1.tif; Fig3.tif)

Additional Modifications Made While Proofreading and Double-Checking Citations

In addition to the changes mentioned above, we have made the following additional modifications:

1. The phrase, “Relative to targeted sampling, complete sampling had 32.05 fewer infections at the under boundary under 8 contacts/day…”, has been corrected to say, “Relative to targeted sampling, complete sampling had 32.05 fewer infections at the upper boundary under 8 contacts/day…” (pg. 19, line 403). 

2. We previously stated that Cas et al. (2021) discussed Furman University’s COVID-19 response. Although this is implied within the paper, the authors only state that they are describing the response of a college in South Carolina. We have therefore removed Furman University’s name to follow this convention (pg. 27, line 616; pg. 32, line 724).

3. We have updated our description of Muller and Muller’s (2021) results on pg. 31, Lines 696-697 to more accurately reflect the authors’ findings on the effectiveness of contact tracing relative to surveillance testing with complete or nearly complete sampling. 

The description previously read, “This intervention has a long and proven track record in public health policy, translates well to novel diseases, and was found to have a comparable effectiveness to surveillance testing with complete sampling in other modeling studies of COVID-19 (41).” 

The description now reads, “This intervention has a long and proven track record in public health policy, translates well to novel diseases, and was found to have a comparable effectiveness to surveillance testing with complete or nearly complete sampling in other modeling studies of COVID-19, at least under certain measures (44).”

4. In our S1 Appendix, we previously stated that a CDC seroprevalence study in Connecticut estimated that between 12.82% and 23.26% of all infections led to a reported case. A second review of the paper (Havers et al. 2020) found that it was actually between 12.61% and 22.77% of all infections. This statement has now been corrected (S1 Appendix, pg. 2, line 39).

---

## [Editor Report · Decision Letter 1]

28 Apr 2023

Monitoring and responding to emerging infectious diseases in a university setting: A case study using COVID-19

PONE-D-23-00956R1

Dear Dr. Soda,

We’re pleased to inform you that your manuscript has been judged scientifically suitable for publication and will be formally accepted for publication once it meets all outstanding technical requirements.

Kind regards,

Barbara T Rumain, PhD

Academic Editor

PLOS ONE

---

## [Editor Report · Acceptance letter]

9 May 2023

PONE-D-23-00956R1 

Monitoring and responding to emerging infectious diseases in a university setting: A case study using COVID-19 

Dear Dr. Soda:

I'm pleased to inform you that your manuscript has been deemed suitable for publication in PLOS ONE. Congratulations! Your manuscript is now with our production department. 

Kind regards, 

on behalf of

Dr. Barbara T Rumain 

Academic Editor

PLOS ONE